# Testing the Effectiveness of the “Smelly” Elephant Repellent in Controlled Experiments in Semi-Captive Asian and African Savanna Elephants

**DOI:** 10.3390/ani13213334

**Published:** 2023-10-26

**Authors:** Marion R. Robertson, Lisa J. Olivier, John Roberts, Laddawan Yonthantham, Constance Banda, Innocent B. N’gombwa, Rachel Dale, Lydia N. Tiller

**Affiliations:** 1WildAid, 220 Montgomery Street #1200, San Francisco, CA 94104, USA; 2Game Rangers International, Plot 2374, The Village, Leopards Hill Road, Lusaka 10101, Zambia; lisa@gamerangersinternational.org (L.J.O.); constance@gamerangersinternational.org (C.B.); 3Golden Triangle Asian Elephant Foundation, 229 Moo 1, Chiang Saen, Chiang Rai 57150, Thailand; jroberts@anantara.com (J.R.); laddawan.yon@gmail.com (L.Y.); 4Department of National Parks and Wildlife, Ministry of Tourism, Chilanga 10101, Zambia; innobilly86@yahoo.com; 5Department for Psychosomatic Medicine and Psychotherapy, University for Continuing Education Krems, 3500 Krems an der Donau, Austria; racheldale07@gmail.com; 6Amboseli Trust for Elephants, Langata, Nairobi 15135, Kenya; lydiatiller@gmail.com; 7Durrell Institute of Conservation and Ecology, University of Kent, Canterbury CT2 7NZ, UK

**Keywords:** human–wildlife conflict, crop-raiding, olfaction, mitigation methods, elephant repellent, elephant personality, animal captivity

## Abstract

**Simple Summary:**

Mitigating and reducing the impacts of elephant crop-raiding has become a major focus of conservation intervention. By observing the behaviour amongst two groups of semi-captive African and Asian elephants in Zambia and Thailand, we found that a novel olfactory crop-raiding mitigation method called the “smelly elephant repellent” elicited clear reactions from the elephants. However, unlike trials with wild elephants, the repellent did not prevent the elephants from entering areas or eating food protected by the solution. We found that elephant personality played a role in responses towards the repellent, as the individuals that entered the experimental plots were bolder and more curious individuals. Although captive environments provide controlled settings for experimental testing, the ecological validity of testing human–elephant conflict mitigation methods with captive wildlife should be strongly considered. Understanding animal behaviour is essential for improving human–elephant coexistence and for designing deterrence mechanisms, and the smelly elephant repellent may be a useful mitigation method when used in combination with other methods.

**Abstract:**

Crop-raiding by elephants is one of the most prevalent forms of human–elephant conflict and is increasing with the spread of agriculture into wildlife range areas. As the magnitude of conflicts between people and elephants increases across Africa and Asia, mitigating and reducing the impacts of elephant crop-raiding has become a major focus of conservation intervention. In this study, we tested the responses of semi-captive elephants to the “smelly” elephant repellent, a novel olfactory crop-raiding mitigation method. At two trial sites, in Zambia and Thailand, African elephants (*Loxodonta africana*) and Asian elephants (*Elephas maximus*) were exposed to the repellent, in order to test whether or not they entered an area protected by the repellent and whether they ate the food provided. The repellent elicited clear reactions from both study groups of elephants compared to control conditions. Generalised linear models revealed that the elephants were more alert, sniffed more, and vocalised more when they encountered the repellent. Although the repellent triggered a response, it did not prevent elephants from entering plots protected by the repellent or from eating crops, unlike in trials conducted with wild elephants. Personality played a role in responses towards the repellent, as the elephants that entered the experimental plots were bolder and more curious individuals. We conclude that, although captive environments provide controlled settings for experimental testing, the ecological validity of testing human–elephant conflict mitigation methods with captive wildlife should be strongly considered. This study also shows that understanding animal behaviour is essential for improving human–elephant coexistence and for designing deterrence mechanisms. Appreciating personality traits in elephants, especially amongst “problem” elephants who have a greater propensity to crop raid, could lead to the design of new mitigation methods designed to target these individuals.

## 1. Introduction

Human expansion and habitat conversion have led to a decline in African and Asian elephant ecosystems [1,2]. This land use change has increased interactions and resource competition between humans and elephants [3]. The resultant human–elephant conflict is one of the greatest challenges facing conservation today, due to the significant costs incurred for both people and elephants [4,5,6].

Human–elephant conflict can have negative impacts on people through loss of crops, damage to property, loss of livestock, and, in some cases, human injury or loss of life [6,7]. This can have detrimental impacts on the food security of already impoverished farmers and may be increasingly compounded by climate change [8]. Indirect costs can also occur, including restriction of movements, disruption to daily routines, fear, and sleepless nights [9]. These impacts can play a significant role in eroding tolerance towards elephants, leading to retaliatory killing [10,11], impacting elephant populations and hampering wildlife conservation efforts [12,13].

Crop-raiding by elephants is one of the most prevalent forms of human–elephant conflict and is increasing with the spread of agriculture into wildlife range areas [14,15]. It is well documented that, at certain times of the year, the nutritional value of cultivated food crops is greater than that of natural forage [16,17,18,19], which is attractive to elephants [20]. However, crop-raiding poses a significant risk to elephants through attacks from farmers [11], and so elephants must weigh up the risk versus the nutritional gain from the crops. Due to this high-risk, high-gain strategy, in some areas, only male elephants have been reported to crop-raid [18,19,21,22]. This risk-taking behaviour of males could have evolved as a result of strong sexual selection for large body size and condition-dependent mating success in males [23]. Crop-raiding can lead to gains in body size for elephants, as cultivated food crops are highly nutritious [23]. Risk assessment behaviour has been documented in elephants around humans, as they move faster through dangerous areas where there is a risk of mortality [24,25], and they move at night when there is less risk [25,26].

As the magnitude of conflicts between people and elephants increases across Africa and Asia, mitigating and reducing the impacts of elephant crop-raiding has become a major focus of conservation intervention. A variety of short-term solutions have been developed, which include physical, visual, and acoustic interventions that deter elephants from entering farms and/or eating crops. Traditional, low-cost strategies include burning fires, using dogs to alert to elephant presence, and noise making, such as banging pots and pans and shouting [14]. More recently developed methods include wire, electric, chilli and honeybee fencing [14], metal strip fences [27], trenches, buffer cropping, sirens, predator playbacks, strobe lighting [28], soft virtual boundaries [29], and the smelly elephant repellent [30]. These strategies are often selected by farmers based on their affordability, accessibility, and traditional effectiveness [14]. However, effectiveness varies widely [14,31] and may depend on the landscape and variation in elephant behaviour and cognition [32]. This is of particular concern, as some individual elephants are learning to circumvent conflict mitigation strategies such as electric fences [33,34]. Given the high cognitive capabilities of elephants, including their ability to alter their behaviour in response to risk, understanding elephant behaviour is key to designing mitigation methods [35].

By taking into account an elephant’s perspective, we may be able to design more effective mitigation methods [35]. For example, elephants have an excellent sense of smell [36] and use olfaction to navigate their world, for foraging, and for social decision making [37,38]. The knowledge of this key sensory perception has led to more olfactory-based mitigation methods being designed [39]. These include: (1) using scent to mask the smell of ripening crops [40]; (2) using the chemical compound capsaicin in chilli, as chilli can cause irritation to elephants’ eyes and noses and stimulate olfactory receptors [41,42]; and (3) using a smell that elephants are averse to, e.g., bee pheromones [43] or predator scents [44].

The use of chilli pepper (*Capsicum* spp.) is one of the most widely tested olfactory-based mitigation methods. Various studies have trialled chilli-based methods, including fences of chilli oil-soaked cloths and chilli briquettes [41,45,46,47]. Chilli has been effective in some contexts (e.g., [48]), but in others has shown a low efficiency when compared to easier and cheaper methods such as community guarding [49]. Despite the positive impact that chilli can play in mitigating elephant damage to crops, the long-term sustainability of this method has been questioned due to the difficulty, expense, and labour required for its application.

Another olfactory-based mitigation method that includes chilli in its ingredients is the “smelly” elephant repellent, also known as the “smelly repellent” or just “repellent”, which is a novel foul-smelling liquid made of common natural ingredients [50]. The ingredients include chilli, garlic, ginger, neem leaves, cooking oil, dung, and rotten eggs. After the preparation of the solution, the mixture is left to mature for a strong, unpleasant odour to develop. This method was developed in Uganda and initially showed positive results as a domestic animal repellent, before being utilised as an olfactory deterrent for elephants. In trials with wild elephants, the smelly elephant repellent was sprayed directly onto crops or put into perforated bottles hung on a rudimentary fence around a crop field.

The smelly repellent showed high levels of effectiveness in trials on 40 farms in Uganda and Kenya. In Uganda, 82% of 309 attempted elephant crop-raiding incidents recorded at 30 farms on the northern boundary of Murchison Falls National Park were deterred. That is to say that elephants approached the farms and could have crop-raided, but were sufficiently put off by the smelly repellent so as to retreat without eating. In Kenya, the repellent deterred 63% of 24 attempted elephant crop-raiding incidents at 10 farms in the Tsavo Conservation Area, and there was a significant effect of the repellent on test sites compared to control sites [30]. The study highlighted the potential for the repellent to be a helpful crop-raiding mitigation tool for farmers, as the community also responded positively to using it. Moreover, the repellent is relatively cheap and quick to produce from ingredients readily available in most countries that elephants inhabit. It is important that mitigation methods are cheap, effective, and have community buy-in, as without this, uptake will not be successful [14,45,51,52].

Following the trials with wild African elephants, the next step was to test the smelly repellent on Asian elephants to determine the effectiveness and potential for use of this method in Asia. Given the caveats and challenges of conducting trials in the wild, such as the length of time required to gather statistically sound datasets and the impossibility of ruling out other deterring factors, we decided to carry out the trials with semi-captive elephants. Here, it was easier to control the environment and be able to fully observe the elephant behaviour when exposed to the repellent. Utilising two groups of semi-captive elephants, one in Asia and one in Africa, also gave us the chance to determine whether reactions to the repellent were species reactions or captive elephant reactions. Furthermore, as described above, individual learning and behaviour may also affect crop-raiding behaviours, and it is important to consider this for mitigation efforts [29,35]. Therefore, a captive setting also allowed for the testing of individual elephant traits, which may contribute to crop-raiding.

In this study, we aimed to determine the effectiveness of the smelly elephant repellent as an elephant crop-raiding deterrent. We presented semi-captive African and Asian elephants with a foraging opportunity with either the smelly repellent or a control condition (water) and recorded their behavioural responses. Additionally, personality traits were measured from each individual and compared with their crop-raiding behaviours. Based on a previous study with wild elephants [30], we predicted that elephants would be more likely to show a behavioural response, less likely to enter the plot, and less likely to eat in the repellent trials than in the control trials. We also predicted that personality would have an effect on likelihood to enter the plot and likelihood to eat the food in the repellent trials, but this was explorative to better understand these little-researched effects, and so we did not make specific predictions regarding the directionality of the results.

## 2. Materials and Methods

### 2.1. Study Sites

The smelly elephant repellent was tested at two sites between February and September 2021.

#### 2.1.1. Thailand

The first site was at the Golden Triangle Asian Elephant Foundation (GTAEF), located in Chiang Saen in northern Thailand. The facility was home to 26 elephants that were rescued from the streets of Bangkok, the logging industry, or had been transferred from tourist trekking camps. The elephants are used as part of the elephant camp programmes at the Anantara Golden Triangle Elephant Camp and Resort and the Four Seasons Golden Triangle Tented Camp. The elephants are provided with access to natural habitat at all times, with artificial shelter provided during extreme weather or veterinary observation. They are fed from four to seven times a day on natural grasses and fruits and bathed two or three times a day. The elephant’s mahout (the daily caretaker who is often also the elephant’s owner), two full-time staff veterinarians, and senior management provide daily care and ensure that proper elephant welfare practice is in place. Details of the elephants that participated in this study can be found in Appendix B (Table A1).

#### 2.1.2. Zambia

The second test site was at the Game Rangers International’s (GRI) Kafue Release Facility, located in Kafue National Park, western Zambia. GRI works in close partnership with the Department of National Parks and Wildlife (DNPW) to empower rangers and communities to conserve nature through natural resource protection, community outreach, and wildlife rescue. The wildlife rescue programme focuses on the rescue, rehabilitation, and release of elephants orphaned as a result of human–elephant conflict and poaching. The release facility at Kafue is the final stage of rehabilitation, where the aim is to provide the elephants with the most natural environment possible, in order that, over time, they gradually become competent and confident enough to live in the wild without the support of humans. The younger elephants spend their nights in a predator-proof area with shelter to ensure their safety. Once elephants are considered physically capable of defending themselves against predators, they are encouraged to remain outside of the outer boma (enclosure). Elephants are only encouraged to the outer boma for two hours over lunchtime, where they have access to pellets (for additional nutrition), have pools for drinking, and mud-bathing areas. The elephants are escorted on daily walks in the national park by elephant keepers and armed wildlife rangers. To prepare them for a life in the wild, during these walks, the orphaned elephants decide where they walk and feed, and if they interact with wild elephants. Details of the elephants that participated in this study can be found in Appendix B (Table A2).

### 2.2. Smelly Elephant Repellent

The smelly elephant repellent is made of a mixture of natural ingredients that are cooked together, matured, and then strained, producing a potent smelly liquid. The ingredients (chilli, garlic, ginger, cooking oil, eggs, neem leaves, and dung) were purchased from nearby towns or collected locally, and were pummelled before being cooked together and left to mature in sealed containers for approximately four weeks [50]. The production process of the repellent for both studies was conducted on-site by the GTAEF and GRI teams, and a previous study found that the scent of the repellent lasted for around 7–8 weeks before fading in wild conditions [30].

### 2.3. Trials and Data Collection

#### 2.3.1. Thailand

In Thailand, we replicated a typical “farm” set up, using a 10 × 10 m plot of land in the grasslands of the organisation’s elephant camp. The “crop” was maize cobs or pieces of sugarcane distributed at roughly 1 m intervals around the edge of the plot (reachable from outside the fence in the fence conditions), another cob or cane layer further in, and at least two large piles of maize or sugarcane in the centre of the field. This distribution was to make the crop obvious to the elephants and replicate their normal feeding context. The plot was used for four different test and control conditions (Figure 1):Repellent fence: The test fence with the repellent in plastic bottles. A fence line using rope was erected using 4–8 poles. Plastic bottles were hung onto the fence line at a distance of 1 m apart. The bottles were filled to roughly a quarter full with repellent and had holes punctured in them to allow the smell of the repellent to diffuse.Water fence: The control fence with water in plastic bottles. The same process as above was applied, however, the bottles were quarter filled with water instead of repellent.Repellent spray: The test crops were sprayed with repellent. The fence line around the plot was removed and the crops in the plot were dipped in the repellent. The ground was not sprayed so that the same plot could be used for each scenario.Water spray: The control crops were sprayed with water. The crops in the unfenced plot were dipped in water.

**Figure 1 animals-13-03334-f001:**
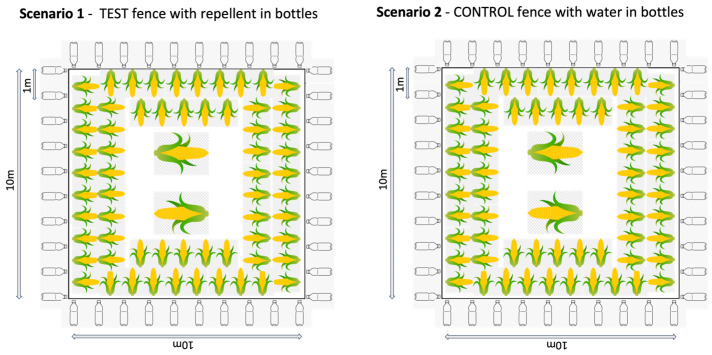
Set up of the four different trial situations in Thailand.

For these trials, 13 female elephants in five groups of two to three individuals, based on long-term social bonds, were tested. Prior to testing, each elephant group was taken to the plot without any repellent or water in order for them to learn that food was available in this location. Each group was tested on each condition four times, meaning that the total number of trials across the different conditions was 12 for each elephant. The elephants were only tested on one condition per day, and the order of the condition type order was randomised.

Each trial was a maximum of 30 min long, and for each trial, the elephants were released into the grassland approximately 30 m from the plot. Distance markers were set up to allow the observer to measure how close the elephants got to the plot. An observer filmed the trial from behind the plot to record the reactions of the elephants to the fence/sprayed crops. The observer also recorded the following: (1) the behavioural reaction of the elephants to the plot (see below); (2) whether the elephants broke through the fence and entered the plot; and (3) whether the elephants ate the food. We observed and recorded the following behaviours: 1. Sniffing, 2. Alertness, 3. Vocalising, 4. Approaching the plot straight away, 5. Turning away from the plot immediately, 6. Strong reaction, 7. Did nothing, and 8. Any other behaviour of note. See Table A3 in the Appendix C for behaviour definitions.

#### 2.3.2. Zambia

In Zambia, a different set up was used to fit the conditions of the facility. The orphaned elephants return from the bush to their boma (enclosure) every lunchtime and evening. Their diet is supplemented with browse and elephant pellets when they return, and this is usually spread out throughout the boma in different places every day so that the elephants learn to use their sense of smell to find it. We replicated a farm scenario using a 40 × 40 m plot of land inside the elephant boma, with the “crop” being pellets. The same number of cups of pellets as the elephants would usually have access to (*n* = 15) were placed inside the “farm” test area in the boma. A “no choice” condition, where pellets were only provided inside the test area, and not elsewhere in the boma, was also tested. The intention here was to create a highly motivating condition, much like crop-raiding in the wild, and so putting the repellent to an even stronger test. The 40 × 40 m farm plot was used for four different test and control conditions (Figure 2):Repellent choice: The test pellets were sprayed with repellent and there was the usual choice of browse/pellets in the boma.Water choice: The control pellets were sprayed with water and there was the usual choice of browse/pellets in the boma.Repellent no-choice: The test pellets were sprayed with repellent and there was no choice of browse/pellets in the boma.Water no-choice: The control pellets were sprayed with water and there was no choice of browse/pellets in the boma.

**Figure 2 animals-13-03334-f002:**
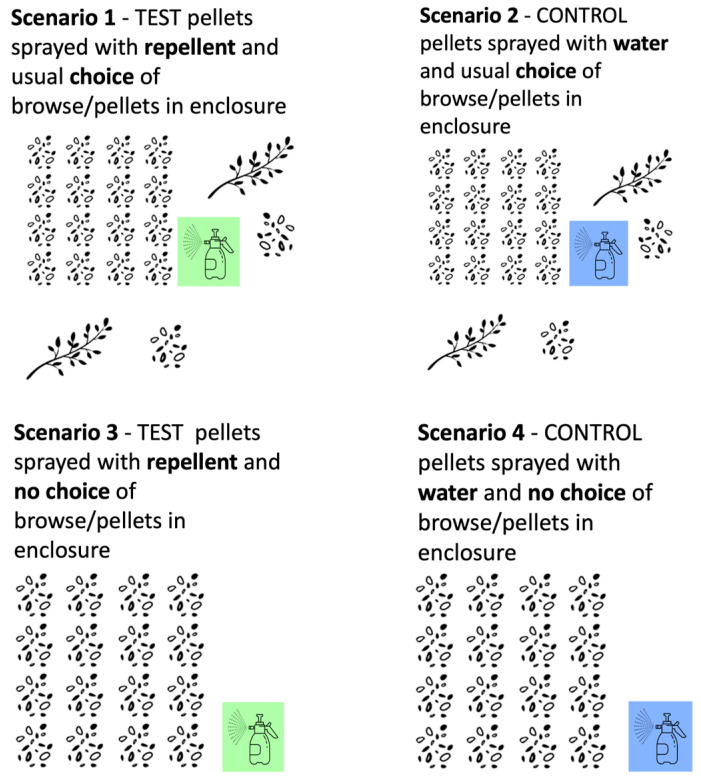
Set up of the four different trial situations in Zambia.

For each trial, up to 13 elephants entered the boma approximately 160 m from the test/control area. Marks on the ground allowed the observers to measure how close the elephants got to the 40 × 40 m farm plot. The elephants were given the entire duration that they were in the boma—approximately 2 h—for each trial. An observer filmed each session from the research tower, GoPro cameras were placed along the fence, still photographs were captured, and a minimum of two observers filled out a sheet recording the different reactions of the elephants to the pellets, specifically recording the same factors as those in the Thailand trial.

#### 2.3.3. Personality Assessment

Elephant personality was assessed with the Elephant Personality Survey [53]. Two raters from each site, who had been working with the elephants for at least three years (Thailand: five and 23 years; Zambia: three and 14 years), rated the traits for each of the elephants across three domains: interactions with the physical environment; interactions with other elephants, and interactions with humans. Each trait was scored on a seven-point Likert scale (e.g., extremely shy, quite shy, slightly shy, neutral, slightly bold, quite bold, and extremely bold). We then selected seven traits to analyse in comparison with the behaviour in the trials: interactions with the physical environment—curiosity, confidence, playfulness, and fearless/timid, and interactions with humans—curiosity, aggressiveness, and shy/bold. These traits were chosen because studies have shown that bolder individuals are more likely to engage in human–animal conflict [54,55]. The scores from the Zambia sample were less varied than those from the Thai sample, and so to rule out rater effects, two further raters completed the survey and inter-rater reliability was assessed across all four raters for this site. Inter-rater reliability was assessed with the intraclass correlation coefficient using the irr package in R. Significant reliability was reached across all traits in both sites (*p*-values < 0.05), with ICC values ranging from 0.4 to 1.0. When the scores for an individual differed, the mean score was taken for the analyses.

#### 2.3.4. Data Coding and Statistical Analysis

In Thailand, the data were collected live during the session by one observer, and 48% of the videos (*n* = 33) were also coded by a second coder (R.D.). In Zambia, the data were collected live during the session by two coders. The behavioural responses during the trials were coded categorically (yes/no) for occurrence across the whole trial.

To analyse the effects of condition and personality on the various behaviours, generalised linear models were run using the ‘glmer’ function in the lme4 package in R version 4.1.2 with the binomial family and backward elimination [56]. The full models were compared to the null models, without condition or personality included, to assess the effects of condition/personality on each dependent variable. The following construct depicts the basic model used for all the GLMMs:Response_ijk_ ~ condition_i_ + animal_ij_ + e_ijk_

In this model example, “condition_i_” is the fixed categorical effect of condition, “animal_ij_” is the random animal (or animal within group for Thailand) effect with mean zero, and “e_ijk_” is the random residual with mean zero.

For assessing personality traits, only data from the repellent trials were used and the models were split into two groups, such that the traits related to interactions with the physical environment were analysed together and interactions with humans were analysed together. For the personality models, sex was also included as a factor for the Zambia sample to control for this (in Thailand all elephants were female). For all the models, elephant ID and group were included as random factors for Thailand and only elephant ID was included as a random factor for the Zambia sample, as all individuals were part of the same group.

## 3. Results

### 3.1. Results

#### 3.1.1. Thailand

In Thailand, *n* = 13 elephants participated in the trials (Appendix C, Table A3), with each trial lasting for 30 min. The results for the effects of the repellent fence, water fence, repellent spray, and water spray conditions on the elephant behaviour in the Thailand trial are presented in Table 1.

Higher likelihoods of sniffing (e.g., spray condition: 60% more likely), alertness (spray: 21% more likely), and vocalisations (spray: 31% more likely) were observed in the two repellent conditions compared to the two water conditions. The elephants were also less likely to approach the plot immediately and approached the plot and then turned away more often in the repellent conditions as compared to the water conditions. However, the repellent did not affect whether they ultimately entered the plot over the course of the whole session, but having a fence did, with the elephants being slightly less likely to enter in the two fence conditions than the spray conditions (e.g., 20% less likely to enter with the repellent fence than repellent spray). Having said that, the repellent spray did somewhat deter them from eating after entering (26% less likely to eat in repellent spray than water spray).

#### 3.1.2. Zambia

In Zambia, *n* = 15 elephants participated in the trials (Appendix B, Table A2), with each trial lasting approximately two hours. The results for the effects of the repellent choice, repellent no-choice, water choice, and water no-choice conditions on the elephant behaviour in the Zambia trials are presented in Table 2.

The elephants in Zambia were more likely to sniff in the repellent no-choice condition (25% more likely than water no choice) and more likely to eat in the water no-choice condition (26% more likely than repellent no choice). There was an effect of condition on entering the plot, but this seemed to be more about whether they had choice or not, not whether there was repellent.

There was no effect of condition on alert behaviours, vocalisations, and approaching and turning away. Turning away immediately and strong reactions happened too rarely to analyse statistically, but it is notable that a strong reaction only occurred in repellent conditions.

### 3.2. Effect of Personality

#### 3.2.1. Thailand

Generalised linear models revealed that elephants with a higher playfulness in the physical environment were significantly more likely to enter the plot in repellent trials (z = 2.05, *p* = 0.04; Figure 3). There were no effects of curiosity, fearfulness, or confidence in the physical environment on entering the plot (curiosity: z = −0.1, *p* = 0.92, fear: z = −0.83, *p* = 0.41, and confidence: z = 1.45, *p* = 0.15) or eating (curiosity: z = 1.57, *p* = 0.12, fear: z = −0.78, *p* = 0.44, and confidence: z = 1.00, *p* = 0.32), nor an effect of playfulness on likelihood to eat the food (z = −0.09, *p* = 0.93).

Regarding behaviour around humans, those rated as bolder around humans, although not statistically significant, did show a tendency to be more likely to enter the plot in repellent trials (z = 1.87, *p* = 0.06; Figure 4). There were no effects of aggressiveness or curiosity towards humans on likelihood to enter the plot (aggression: z = 0.07, *p* = 0.94, curiosity: z = −0.60, *p* = 0.55) or eat the food (aggression: z = 0.33, *p* = 0.74, curiosity: z = −0.18, *p* = 0.86), nor an effect of boldness on eating the food (z = 1.54, *p* = 0.12).

#### 3.2.2. Zambia

Generalised linear models revealed that, in repellent trials, elephants with a higher playfulness in the physical environment had a tendency to be more likely to enter the plot (z = 1.67, *p* = 0.09). There were no effects of elephant sex, confidence, fear, or curiosity in the physical environment on entering the plot (sex: z = 0.24, *p* = 0.81, confidence: z = −0.52, *p* = 0.60, fear: z = 0.42, *p* = 0.67, and curiosity: z = −0.37, *p* = 0.71) or eating (sex: z = 1.14, *p* = 0.25, confidence: z = −0.59, *p* = 0.56, fear: z = 0.70, *p* = 0.48, and curiosity: z = −0.17, *p* = 0.87), nor an effect of playfulness on likelihood to eat (z = 0.13, *p* = 0.89).

In terms of personality with humans, elephants rated as more bold with humans were more likely to enter the plot than shy elephants (z = 2.04, *p* = 0.04; Figure 5) and there was a trend for elephants more curious around humans to be more likely to enter (z = 1.69, *p* = 0.09), but no effect of aggressiveness (z = −0.23, *p* = 0.82) or sex (z = −0.12, *p* = 0.90). Interestingly, boldness did not affect likelihood to eat the food (z = −0.15, *p* = 0.88) and nor did sex (z = 0.33, *p* = 0.74), but elephants more aggressive or more curious towards humans were significantly more likely to eat in repellent trials (aggressive: z = 96.44, *p* < 0.0001; Figure 6, curiosity: z = 25.25, *p* < 0.0001).

## 4. Discussion

The smelly elephant repellent triggered behavioural responses in the elephants in both Thailand and Zambia, indicating that the individuals were able to detect the repellent. However, the repellent did not prevent all the elephants from entering the plot (in Thailand) and/or eating the food sprayed with the repellent (in Thailand and Zambia). In Thailand, the fence deterred the elephants from entering the test plot as, when there was no fence present (during the spray conditions), the elephants entered the test plot more. In both Thailand and Zambia, even though the elephants ate the food during the repellent spray trials, they were less likely to eat compared to when the food was sprayed with water. In both species, the elephants behaviourally reacted more towards the repellent conditions compared to the water conditions, as they exhibited enhanced sniffing, alertness, and vocalisation behaviours. Anecdotal evidence from the observers suggests a dislike for the repellent, as some of the elephants were seen to wipe the repellent off the food (see Appendix A). This was not coded systematically in our observations, as we were not expecting to see this behaviour, and so it could have occurred more than was noted.

The results of this study, showing the elephants to enter the plot and eat the food, were surprising, as the smelly elephant repellent has successfully deterred wild elephants from entering crop fields in Kenya and Uganda [30]. However, this type of result has been documented previously when trialling the use of beehive fences as a human–elephant conflict mitigation deterrent. In the wild, elephants were deterred from entering farms due to beehive fences [57,58], but, with captive elephants, the beehives had no impact [59]. Thus, although captive environments provide controlled settings for experimental testing, the ecological validity of testing conflict mitigation methods with captive wildlife should be strongly considered [38,59,60,61]. This may be due to differences between captive and wild elephants in social and physical conditions, stress levels, and interactions with or exposure to humans.

The presence of a mahout/carer during the trials in both Thailand and Zambia could have impacted the elephants’ performances [62]. The working performance of captive elephants during novel situations has been documented to improve with mahout/carer presence. For example, in Myanmar, captive elephants were more likely to cross an unfamiliar barrier in the presence of their mahouts, and more so if they had known their mahout for over a year [62]. This presence may have made the elephants in this study feel more relaxed and safe to enter the unfamiliar test plot or eat the novel food. In Zambia, a feeling of safety was also previously anecdotally observed when spiky stones were being tested as a mitigation method and one of the elephants injured itself by walking straight over them, indicating that the elephant did not see the spikes as a threat (L. J. Olivier, pers. comm.). The captive setting in this study is very different to real life, despite our best attempts to recreate a real-world scenario. In the wild, entering a field of crops and consuming crops poses a significant risk to the survival of elephants through retaliatory injury by farmers [11]. However, the captive elephants in this study were not faced with this kind of threat and so there was no real risk for them to enter the test plot. In addition, captive elephants, in general, are not kept in natural social groups, which could impact their behaviour [35]. Thus, the lack of socio-ecological validity makes it very difficult to test behaviours associated with the conservation issues facing wild populations [35].

Finally, in captivity, individuals have more exposure to man-made objects and may be more used to human smells compared to wild individuals. This may result in less neophobia among captive animals compared to wild animals when exposed to a novel object and smell, like the set up in this experiment [63,64]. Nonetheless, behavioural effects of the repellent were still observed in our semi-captive samples, indicating the detection of and reaction to the repellent.

Despite previous findings that male elephants are more likely to crop-raid [18,19,21,22], our results in the Zambia sample showed that sex did not have a significant effect on behaviour when included in the same model as personality. Therefore, it seems that, in this sample, personality was more of a driver of behaviour around the repellent than sex. However, it is worth noting that in Zambia the elephants were younger, and so this result cannot be generalised to adults.

Personality differences between individuals, such as boldness towards novel scenarios and the ability to thrive in stressful captive environments, can impact performance in different situations [65,66]. This was seen in our study, as personality had an impact on whether the elephants entered the test plot or ate the food in the repellent trials. Both the Thai and Zambian elephants, two different species, showed similar effects of personality traits on repellent interactions, such as boldness and playfulness, with the bolder, more playful elephants being more likely to enter the plot. Boldness is related to the probability of exploring new things and neophilia, with highly neophilic animals being quick to approach and explore a novel object, while highly neophobic animals are slow to do so [60,67]. Wild animals are more neophobic than captive animals [60,61], suggesting that the repellent would be predicted to have a stronger effect on wild animals than on those in captivity.

These personality traits (boldness and playfulness) in wildlife are likely to play a role in determining whether some individuals are more or less likely to engage in risky behaviours that result in human–wildlife conflict [32,33,68]. Bolder animals are more likely to take risks. For example, bolder wild house sparrows are more likely to invade new areas and explore human-made objects [54]. Bolder coyotes are less likely to be scared by mitigation devices used to protect food [55]. It is likely that these traits will determine whether elephants make the decision to take considerable risks by entering farms and engaging with people by crop-raiding and breaking fences, and why other individuals avoid them by remaining inside protected areas [32,33,68]. Often, these risk-taking elephants are labelled as “problem” animals. However, depending on location, a small percentage of actual populations are “problem” individuals.

## 5. Conclusions

Understanding animal behaviour is essential for improving human–elephant coexistence and designing deterrence mechanisms. Appreciating personality traits in elephants, especially amongst “problem” elephants who have a greater propensity to crop-raid, could lead to the design of new mitigation methods designed to target these individuals. Moreover, the use of existing methods, such as the smelly elephant repellent, combined with other methods could be rotated at different times of the year to slow down learning and habituation [35].

Overall, both the African and Asian elephants in this study showed behavioural reactions to the smelly elephant repellent, and this was further affected by personality traits. Due to the potential differences between captive and wild elephants, caution has to be taken when testing human–elephant conflict mitigation methods, as the results may vary between the two contexts, as is evident when comparing this study to previous findings [30]. These results from captive settings can demonstrate whether elephants respond to the repellent. Future research is needed to understand the effectiveness of the smelly repellent in wild Asian elephants and African forest elephants (*Loxodonta cyclotis*).

## Figures and Tables

**Figure 3 animals-13-03334-f003:**
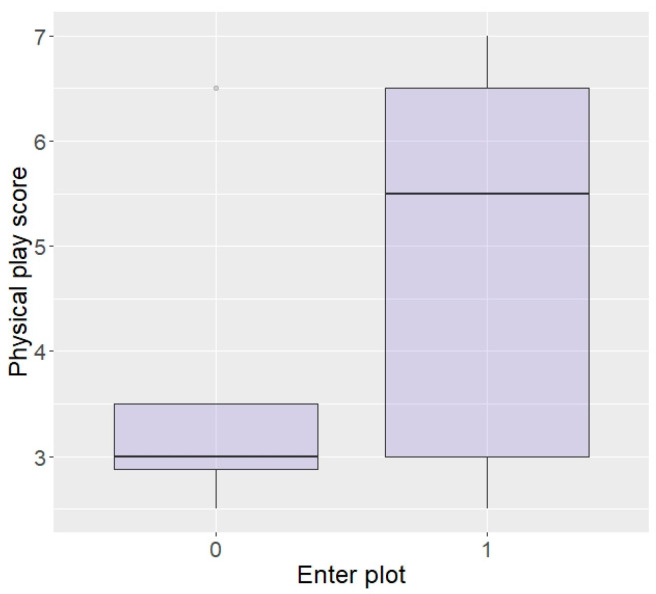
Likelihood of entering the plot according to playfulness in the physical environment (higher score = higher playfulness) in the Thai sample.

**Figure 4 animals-13-03334-f004:**
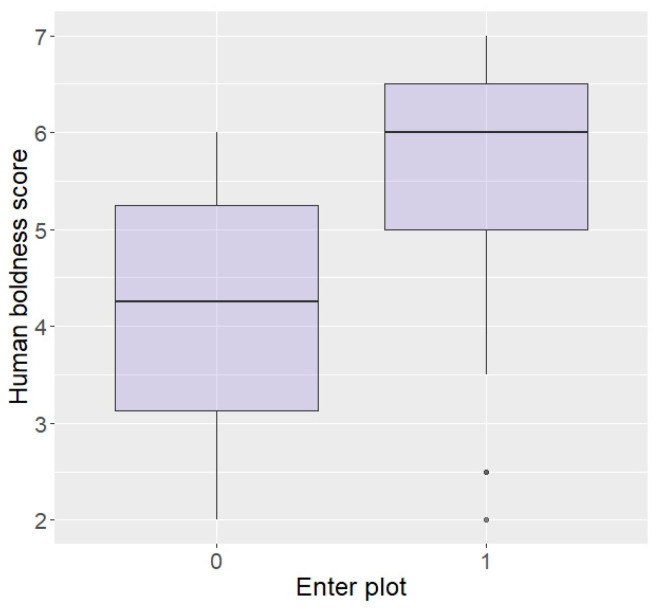
Likelihood of entering the plot according to boldness towards humans (higher score = higher boldness) in the Thai sample.

**Figure 5 animals-13-03334-f005:**
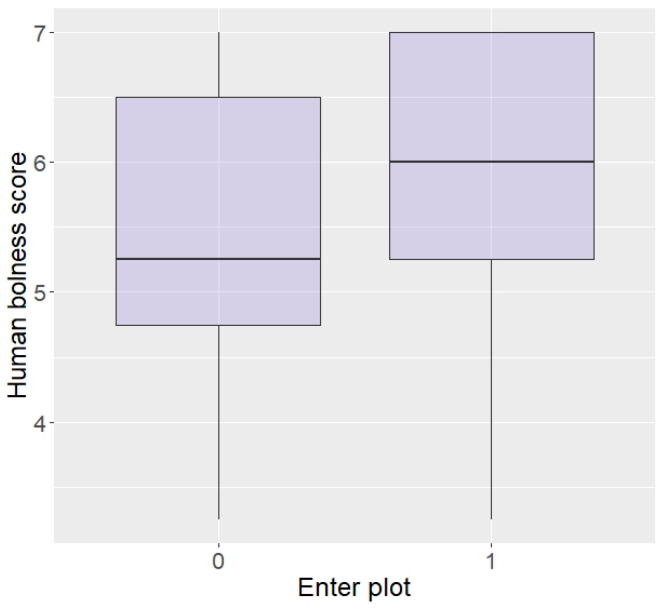
Likelihood of entering the plot according to boldness towards humans (higher score = higher boldness) in the Zambian sample.

**Figure 6 animals-13-03334-f006:**
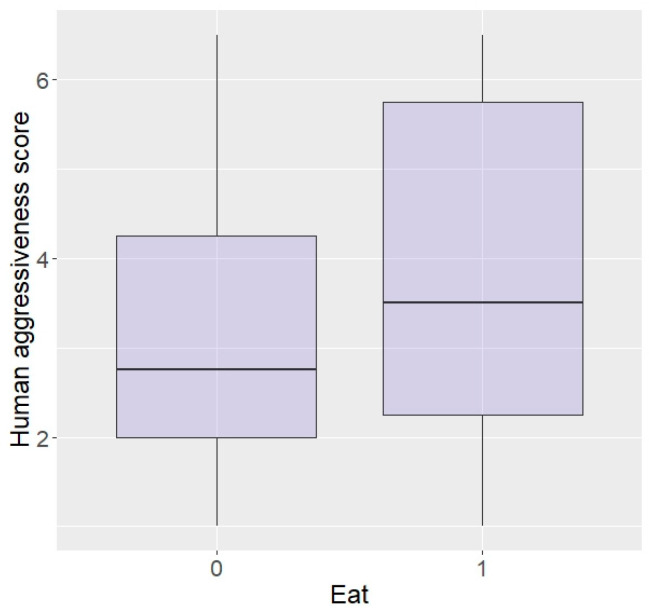
Likelihood of eating the food according to aggressiveness towards humans (higher score = higher aggressiveness) in the Zambian sample.

**Table 1 animals-13-03334-t001:** Percentage of elephants showing each behaviour, according to condition, in Thailand.

Behaviour	Repellent Fence	Water Fence	Repellent Spray	Water Spray	Model *p*-Value
Sniffing	74.36	60.53	97.37	37.50	** *p* ** ** < 0.0001**
Alert	35.90	15.79	21.05	0.00	** *p* ** ** = 0.0001**
Vocalisation	46.15	42.11	50.00	18.75	** *p* ** ** = 0.02**
Approach immediate	66.67	81.58	100.00	100.00	** *p* ** ** < 0.0001**
Approach then turn	58.97	34.21	63.16	31.25	** *p* ** ** = 0.007**
Turn away immediate	5.13	2.63	7.89	0.00	*p* > 0.05
Strong reaction	2.56	2.63	13.16	0.00	** *p* ** ** = 0.04**
Enter plot	79.49	89.47	100.00	100.00	** *p* ** ** < 0.0001**
Eat	82.05	82.86	74.29	100.00	** *p* ** ** = 0.0009**

**Table 2 animals-13-03334-t002:** Percentage of elephants showing each behaviour, according to condition, in Zambia.

Behaviour	Repellent Choice	Repellent No Choice	Water Choice	Water No Choice	Model *p*-Value
Sniffing	53.85	83.02	65.45	58.00	** *p* ** ** = 0.005**
Alert	34.62	26.42	29.09	32.00	*p* > 0.05
Vocalisation	19.23	20.75	14.55	14.00	*p* > 0.05
Approach immediate	36.54	26.42	18.18	36.00	** *p* ** ** = 0.03**
Approach then turn	1.92	1.89	5.45	6.00	*p* > 0.05
Turn away immediate	0.00	3.77	3.64	0.00	-
Strong reaction	0.00	1.89	0.00	0.00	-
Enter plot	29.41	66.04	23.64	48.00	** *p* ** ** < 0.0001**
Eat	21.57	33.96	34.55	60.00	** *p* ** ** < 0.0001**

## Data Availability

The datasets presented in this article can be made available on request directed to the authors.

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
