# Peer review of "Testing the Effectiveness of the “Smelly” Elephant Repellent in Controlled Experiments in Semi-Captive Asian and African Savanna Elephants"

_animals, 2023, doi:10.3390/ani13213334_

Round 1
Reviewer 1 Report
This manuscript focuses on the “smelly elephant deterrent” and the efforts to show its effectiveness as an elephant deterrent for crop raiding/foraging. The study is well designed and focuses on captive animals, though points out the limitations of these types of studies appropriately. This manuscript makes interesting points on how personality may affect behaviors such as crop raiding and is of value. I commend the authors and look forward to its publication. My main critique would be to dig deeper into the literature for appropriate references. There are many studies that came to mind as I read that were not cited in the reference list, especially comparing mitigation types in the wild, history of olfactory deterrents (especially chili-based, as is this method), crop raiding behaviors, and newly tested deterrent methods. I know this can be a challenge for all authors, especially when a study comes out several years after trials occur.
Lines 38 & 479-Scientific names should be italicized with the genus capitalized and species lower case.
2nd paragraph of introduction. There should be more noted here on the dire consequences to food security of already impoverished farmers which is being exacerbated by climate change and other livelihood threats. While HEC is certainly a huge threat for elephants, it can be equally if not more damaging to food security for subsistence farmers.
Line 82. The study referenced here is almost 4 years old. While a great reference there have been other methods pioneered since which you could include. Some examples: strobe lights (Adams et al., 2020), metal strip fences (Von Hagen et al., 2021), the smelly elephant repellant itself (Tiller et al., 2022), and soft virtual boundaries (La Grange et al., 2022). It may be best to provide a reference per deterrent method, dependent on author guidelines. It may also be of value to differentiate in the narrative these more modern fence types from more traditional, low cost (and less effective) deterrents such as guarding, having dogs alert to elephant presence, burning fires, banging pots and pans, sling shots, etc.
Line 121 Could provide more specific references here.
Line 135 This would be the spot to highlight your hypotheses on reactions of elephants to the deterrent or at minimum your predictions.
Line 195 Great job on setting up control plots with bottles with no repellants and sprayed vs. non sprayed fields -this is sometimes not often done properly in wild experiments with controls being only empty fields.
Figure 1. I believe there is a type-o in Scenario 2. I believe you meant to say Control field with NO repellant in bottles or WATER in bottles.
Line 262. Unlike in the Asia elephant trial, the sex and family composition (if any, since they were orphans) of the elephants were not described until the appendix. Given that most crop raiders are adult males, this is a form of bias that needs to be addressed in the discussion section, as most of the Asian trial involved adults and the African trial juveniles, another potential thing that could skew the results-you could discuss any differences in personality/curiosity in younger elephants than adults.
Line 265. In the Asia trial, these were done in 30-minute segments whereas here the time was 2 hours. This also needs to be addressed in the discussion as having more time might have allowed for elephants becoming used to the experiment and becoming more bold than in just the shorter 30 minute trials.
Line 281-This could be an area to reference that males are more often crop raiding individuals. There is also potential to discuss that crop-raiding may be a learned behavior from other males. Patrick Chiyo has many papers discussing male crop raiding behavior but none were referenced in this manuscript (and should be).
Line 320-Here you state that “the repellent spray did deter them from eating after entering” In Table one for the “eat” behavior for the repellent fence, It showed that 82.05 % of elephants did actually eat, though 100% did enter. If the table is correct then this line is confusing since 82% of elephants did in fact eat which is a very high percentage and there is almost no difference between the repellent fence and water fence . I would also put these side by side for clarity (as you do in Table 2), as I would the repellent spray and water spray as there is a much clearer difference and some level of deterrence there. This sentence could be clarified by just stating the percent of elephants that entered and chose to eat. Another question that could use clarification and was not mentioned in the manuscript: Since food was placed within trunk’s reach on the perimeter of the plots, was this considered separately? For example, if an elephant approached the fence and could reach under or over and grab corn to eat, hopefully that would be analyzed differently than if they broke through the fence to eat the corn from the interior. Likely you took this data and it may already be included, it just may need to be described more clearly.
Line 329. Wherever you make statements in your results it is easiest if you give the data rather than the reader having to look it up in the table. e.g. The elephants in Zambia were X% more likely to sniff………..
Lines 352 and 367-In these lines you report p-values over 0.05 as being important. I agree with your assessment here that sometimes values above the normally accepted p-value should be considered and there is often biological relevance. You simply need to state somewhere that you considered p-values of <0.10 and justify it.
Reviewer 2 Report
(abstract)
-underline the italics of scientific names
-repellent life expectancy also should be stated
-statistical tests should be mentioned
(Methods)
(Fig 1, line 187)
-100m2 should be enough to observed the elephant behavior??
(line 294) the model should be presented
(Results)
-the total amount of time should be mentioned for each categories
-what is ‘playfulness’???, ‘fearfulness??’
(in general)
This kind of behavioral studies should be based on amount of time and behaviour should be observed in the equal intervals, and authors did not describe the methods of measuring behaviour in frequency m
-English looks fine and need a minor revision
Reviewer 3 Report
The manuscript submitted for review is interesting and worthy of publication after minor revisions. The authors of the experiment went to a lot of trouble choosing a not easy object for research - elephants. They made sure that the preparation tested on animals (to deter them from cultivation) was based on a recipe of naturally occurring ingredients. So they took pains to make sure that the fragrance mixture repels by smell, but does not have an unduly negative effect on the other senses of the animals tested.
I have a question for the section "Study Sites" the authors write:
"The smelly elephant repellent was tested at two sites between February and September 2021." Did authors show a statistically significant effect of season on the intensity of the effect of this "smelly repellent" on elephants?
Please improve the quality of the lettering on the X and Y axes of Figures 3-6.
Regards
Round 2
Reviewer 2 Report
-Capsicum spp => italic
Most of the comments are properly answered and accept the manuscript as it is.